# Impact of HIV-Related Immune Impairment of Yellow Fever Vaccine Immunogenicity in People Living with HIV—ANRS 12403

**DOI:** 10.3390/vaccines12060578

**Published:** 2024-05-25

**Authors:** Diogo Gama Caetano, Thais Stelzer Toledo, Ana Carolina Souza de Lima, Carmem Beatriz Wagner Giacoia-Gripp, Dalziza Victalina de Almeida, Sheila Maria Barbosa de Lima, Adriana de Souza Azevedo, Michelle Morata, Beatriz Grinsztejn, Sandra Wagner Cardoso, Marcellus Dias da Costa, Luciana Gomes Pedro Brandão, Ana Maria Bispo de Filippis, Daniel Scott-Algara, Lara Esteves Coelho, Fernanda Heloise Côrtes

**Affiliations:** 1Laboratório de AIDS e Imunologia Molecular, Instituto Oswaldo Cruz/Fiocruz, Rio de Janeiro 21040-360, Brazil; diogo.caetano91@gmail.com (D.G.C.); thais_chelsea@hotmail.com (T.S.T.); ana-souza-l@hotmail.com (A.C.S.d.L.); carmembg@ioc.fiocruz.br (C.B.W.G.-G.); dalziza@ioc.fiocruz.br (D.V.d.A.); 2Departamento de Desenvolvimento Experimental e Pré-Clínico (DEDEP), Bio-Manguinhos/Fiocruz, Rio de Janeiro 21040-900, Brazil; smaria@bio.fiocruz.br; 3Laboratório de Análise Imunomolecular (LANIM), Bio-Manguinhos/Fiocruz, Rio de Janeiro 21040-900, Brazil; adriana.soares@bio.fiocruz.br; 4Instituto Nacional de Infectologia Evandro Chagas/Fiocruz, Rio de Janeiro 21040-360, Brazil; michelle.morata@ini.fiocruz.br (M.M.); gbeatriz@ini.fiocruz.br (B.G.); sandra.wagner@ini.fiocruz.br (S.W.C.); lara.coelho@ini.fiocruz.br (L.E.C.); 5Laboratório de Pesquisa em Imunização e Vigilância em Saúde (LIVS), Instituto Nacional de Infectologia Evandro Chagas/Fiocruz, Rio de Janeiro 21040-360, Brazil; marcellus.costa@ini.fiocruz.br (M.D.d.C.); luciana.pedro@ini.fiocruz.br (L.G.P.B.); 6Laboratório de Flavivírus, Instituto Oswaldo Cruz/Fiocruz, Rio de Janeiro 21040-360, Brazil; abispo@ioc.fiocruz.br; 7Lymphocyte Cell Biology Unit, Pasteur Institute, 75015 Paris, France; daniel.scott-algara@pasteur.fr

**Keywords:** yellow fever vaccine, HIV-1, immune activation, immune exhaustion, inflammation, B cells

## Abstract

The yellow fever (YF) vaccine is one of the safest and most effective vaccines currently available. Still, its administration in people living with HIV (PLWH) is limited due to safety concerns and a lack of consensus regarding decreased immunogenicity and long-lasting protection for this population. The mechanisms associated with impaired YF vaccine immunogenicity in PLWH are not fully understood, but the general immune deregulation during HIV infection may play an important role. To assess if HIV infection impacts YF vaccine immunogenicity and if markers of immune deregulation could predict lower immunogenicity, we evaluated the association of YF neutralization antibody (NAb) titers with the pre-vaccination frequency of activated and exhausted T cells, levels of pro-inflammatory cytokines, and frequency of T cells, B cells, and monocyte subsets in PLWH and HIV-negative controls. We observed impaired YF vaccine immunogenicity in PLWH with lower titers of YF-NAbs 30 days after vaccination, mainly in individuals with CD4 count <350 cells/mm^3^. At the baseline, those individuals were characterized by having a higher frequency of activated and exhausted T cells and tissue-like memory B cells. Elevated levels of those markers were also observed in individuals with CD4 count between 500 and 350 cells/mm^3^. We observed a negative correlation between the pre-vaccination level of CD8^+^ T cell exhaustion and CD4^+^ T cell activation with YF-NAb titers at D365 and the pre-vaccination level of IP-10 with YF-NAb titers at D30 and D365. Our results emphasize the impact of immune activation, exhaustion, and inflammation in YF vaccine immunogenicity in PLWH.

## 1. Introduction

Yellow fever (YF) is a sylvatic arboviral disease endemic to the tropical and subtropical regions of Latin America, the Caribbean, and Africa. The disease is a major concern for public health due to the high fatality rate [1,2] and the increasing risk of urban outbreaks associated with the increased circulation of *Aedes aegypti* and the narrowing of the sylvatic borders [3]. In Brazil, urban YF was eradicated in the late 1960s through vector control and vaccination campaigns, but the disease remains restricted to the Amazon region [4], which extends through nine states that house approximately 15% of the country’s total population living with HIV, estimated at around 1 million [5].

YF antiviral-specific treatment is not available, but live attenuated vaccines are available as a primary prevention strategy and widely prescribed to travelers/residents of endemic areas and the general population during outbreaks [3]. During the last decade, wide vaccination campaigns were called in response to outbreaks in 2014–2018, which reached the borders of greatly populated states in the southeast region [6,7]. Together, these states accounted for 162 thousand notified HIV cases in the last decade, representing approximately 40% of HIV infections in Brazil during the historical series [5].

Although safe and highly effective for the general population, precautions are taken while recommending the YF vaccine to immunocompromised individuals due to concerns regarding serious adverse events for live vaccines in those individuals, including people living with HIV (PLWH) [8]. For PLWH, safety and immunogenicity reports are mostly limited to small observational studies and case reports [9,10,11,12,13,14,15], except for a few cohort studies [16,17,18,19,20,21], and indicate that markers of HIV disease progression are associated with a lower level and shorter persistence of YF neutralization antibody (NAb) titers [11,17,18,20]. To address this, a longitudinal prospective non-randomized interventional trial with 280 PLWH was conducted and confirmed the safety of the YF vaccine in PLWH with CD4^+^ cell count ≥200 cells/mm^3^ [22]. However, immunogenicity was impaired in PLWH, particularly in those with a high viral load, low CD4^+^ T cell count, and low CD4/CD8 ratio at vaccination.

Several mechanisms could impact vaccination efficacy in PLWH. Beyond a severe immunodeficiency in advanced stages, HIV infection leads to chronic and generalized immune deregulation that triggers immune activation, affecting several components of the immune response [23,24]. The consequences of this activation state include a shift in T cells towards most differentiated/effector phenotypes [23,25,26,27] and decrease in central memory cells [28,29,30,31]; altered dynamics of monocytes, with an increase in CD16^++^ subsets [32,33,34]; increased frequency of T cells expressing exhaustion markers (e.g., PD-1, TIM-3, TIGIT, and LAG-3) [35,36,37,38,39,40,41,42,43]; and premature aging and immune senescence profiles in cytotoxic T cells [44,45,46,47,48]. In addition, B cell subsets in PLWH present an activation profile similar to T cells, with decreased frequencies of resting memory and increased tissue-like and activated memory subsets [49,50,51,52,53]. Those alterations are associated with disease progression and can be attenuated by antiretroviral treatment, although levels of several markers of immune activation and exhaustion do not fully normalize even after long-term viral suppression [36,54,55,56,57,58]. Furthermore, a study evaluating the response to the YF vaccine in HIV-negative individuals associated the lower level of immune response with an increased level of immune activation and exhaustion in individuals from Uganda [59], highlighting the impact of immune activation in YF vaccine immunogenicity.

To further investigate the mechanisms that could impair YF vaccine immunogenicity in PLWH, our study evaluated the pre-vaccination frequencies of several cellular subsets and inflammation markers that are affected by HIV infection in a longitudinal interventional trial that evaluated the immunogenicity and reactogenicity of the YF vaccine in PLWH and HIV-negative controls [22].

## 2. Materials and Methods

### 2.1. Study Population

This sub-study was conducted within a longitudinal prospective non-randomized interventional trial with PLWH and HIV-negative controls at the Instituto Nacional de Infectologia Evandro Chagas from Fundação Oswaldo Cruz (INI-FIOCRUZ), Rio de Janeiro, Brazil, to evaluate the immunogenicity and safety of a standard single dose of 17DD YFV in PLWH (NCT03132311). The trial design and results have been reported previously [22]. Between May 2017 and May 2018, adults (18 to 59 years old) with no history of prior YF vaccination or disease and no contraindications to the vaccine were eligible for this study [22]. PLWH were required to have a CD4^+^ T cell count >200 cell/mm^3^ at least 6 months before enrolment in this study. The HIV-negative participants were required to have a non-reactive anti-HIV rapid test at enrolment.

At enrolment (D0 visit), all participants received a single standard dose (0.5 mL, subcutaneous) of the 17DD YF vaccine (Bio-Manguinhos - Fiocruz, Rio de Janeiro, Brazil) (21), containing approximately 10^5^ viral particles. Clinical data, including time since HIV diagnosis, time in ART, and nadir CD4 were collected from PLWH. After vaccination, participants were followed on Day 5 and Day 30 and 1 year after enrolment. Blood samples collected at each visit were used for the assessment of CD4^+^ and CD8^+^ T cell counts, quantification of HIV viral load in PLWH, and isolation of peripheral blood mononuclear cells (PBMCs).

### 2.2. Sample Preparation

PBMCs were isolated from whole blood using Histopaque-1077 (Sigma-Aldrich, St. Louis, MO, USA) by density gradient centrifugation, cryopreserved in fetal bovine serum supplemented with 10% DMSO at a concentration of 7–10 × 10^6^ cells/cryovial and stored in liquid nitrogen until use.

### 2.3. CD4^+^ and CD8^+^ T Cell Counts and Plasma Viral Load Quantification

Absolute CD4^+^ and CD8^+^ T cell counts were obtained from whole blood using the MultiTest TruCount-kit and the MultiSet software v3.1x on a FACSCalibur flow cytometer (BD Biosciences, Franklin Lakes, NJ, USA). Plasma HIV-1 viral loads were measured using the Abbott RealTime HIV-1 assay (Abbott Laboratories, Wiesbaden, Germany), with a lower detection limit of 40 copies/mL.

### 2.4. Flow Cytometry

Cryopreserved PBMCs were thawed and rested overnight in RPMI 1640 (Sigma-Aldrich, USA) supplemented with 10% of fetal bovine serum (Gibco, Waltham, MA, USA) at 37 °C, with 5% of CO_2_ and under controlled humidity.

After resting, cells were counted and aliquoted for staining with antibodies for the evaluation of the following cellular populations: naïve, effector, memory, exhausted T cell (Panel 1), senescent T cell (Panel 2), activated T cell, monocyte (Panel 3), peripheral T follicular helper cell (Panel 4), and B cell (Panel 5) subsets. A complete description of the panels and antibodies used in each panel is available in Appendix A.

All the cell samples were stained with FVS450 (BD Biosciences, USA) for dead cell exclusion prior to specific panel staining. After staining, the samples were fixed using PBS-PFA 1% solution and acquired using a BD FACSAria™ IIu flow cytometer (BD Biosciences, USA). Analyses were performed with FlowJo v.10.0.7 (TreeStar, Woodburn, OR, USA).

### 2.5. Plasmatic Markers

The plasmatic levels of IL-4, IL-6, IL-18, IL-21, CXCL10/IP-10, CCL4, and sCD163 were assessed using a custom ProcartaPlex multiplex immunoassay (Invitrogen, Waltham, MA, USA), according to the manufacturer’s instructions, and a MAGPIX reader (Luminex Corp, Austin, TX, USA).

### 2.6. Neutralization Assay

The YF-NAb titers at D30 and D365 were quantified using the micro plaque-reduction neutralization-Horseradish Peroxidase test (μPRN-HRP), carried out at Laboratório de Tecnologia Virológica, Bio-Manguinhos (LATEV, FIOCRUZ-RJ, Brazil), as previously described [60]. The assay values indicate the highest serum dilution tested capable of neutralizing the challenge virus by 50%, with 1:1458 being a higher limit. Titers ≥1:100 (3.15 log10 mIU/mL) were considered reactive and protective, and titers <1:70 were considered non-reactive and non-protective. Samples with results >1:70 and <1:100 were considered inconclusive and reevaluated.

### 2.7. Statistical Analysis

In addition to a group of HIV-negative individuals (HIVneg; *n* = 26), PLWH participants were divided into three groups based on pre-vaccination CD4 counts: CD4 count >500 cells/ mm^3^ (CD4high; *n* = 30), CD4 count between 350 and 499 cells/ mm^3^ (CD4med; n = 20), and CD4 count between 200 and 349 cells/ mm^3^ (CD4low; *n* = 13).

For the descriptive analysis, differences between the groups were verified through the Kruskal–Wallis test for continuous variables and Fisher’s exact test for categorical variables. For the inferential analyses, linear regression fixed-effects models adjusted by age and sex were calculated for each variable evaluated in this study to investigate associations between the main exploratory variable and participant groups. The HIVneg group was considered as a reference category for all models, and *p*-values < 0.05 were considered significant for covariate coefficient estimates. The complete spreadsheet with the models’ statistics is available in Appendix A.

Further, correlation analysis between variables included in this study was evaluated using the Spearman method. Correlations were considered relevant only if *p* < 0.05 and classified as weak (rho between 0.3 and 0.5 or between −0.3 and −0.5), mild (rho between 0.5 and 0.7 or between −0.5 and −0.7), and strong (rho > 0.8 or <−0.8). All statistical analyses were performed in R 4.2.2.

## 3. Results

### 3.1. Clinical and Demographic Characteristics of the Study Groups

A total of 64 PLWH (CD4_high_ [n = 31], CD4_med_ [n = 20], and CD4_low_ [n = 13]) and 27 HIV-negative individuals were included in this study. Table 1 shows the clinical and demographic characteristics of the groups. No significant differences in age and sex were observed between the groups. In PLWH, all individuals were under ART and had an undetectable HIV viral load (<40 copies/mL). There were no significant differences in the time since HIV-1 diagnosis and time since ART initiation, although PLWH with CD4_low_ had lower medians for both variables. In addition, nadir CD4^+^ counts were lower in PLWH with CD4_low_ than in PLWH with CD4_high_ (*p* < 0.001).

### 3.2. Immunophenotyping of T Cell Subsets Commonly Affected by HIV Infection

HIV affects T cell homeostasis, leading to the proliferation of effector, exhausted, and senescent populations, which present limited functionality and may contribute to impaired vaccinal responses. Thus, we sought to evaluate alterations in the subsets in our cohort that could explain differences in YF vaccine immunogenicity.

Based on the data resulting from multivariable linear regression models adjusted by age and sex, we observed increased T cell exhaustion mainly in the CD4_med_ and CD4_low_ groups (Figure 1). Relative to HIV-negative controls, the frequencies of PD-1^+^ cells among total CD4^+^ T cells were higher in the CD4_med_ and CD4_low_ groups (*p* < 0.0001 for both, Figure 1a). The frequency of PD-1^+^ cells among total CD8^+^ T cells was also higher in the CD4_med_ group than in HIV-negative controls (*p* = 0.0028). When evaluating PD-1 expression in T cell subsets, we observed that the population that contributed more to the elevated PD-1 expression among CD4^+^ T cells was TEM cells. Among CD8^+^ T cells, most PD-1^+^ cells were TEM cells.

When evaluating TIGIT as a marker of exhaustion (Figure 1b), adjusted models indicated higher frequencies of TIGIT^+^ cells among all the PLWH groups compared to HIV-negative controls for both total CD4^+^ (*p* = 0.01 for CD4_high_ and *p* < 0.0001 for CD4_med_ and CD4_low_) and total CD8^+^ T cells (*p* = 0.033 for CD4_high_; *p* < 0.0001 for CD4_med_; and *p* = 0.0002 for CD4_low_). As observed for PD-1, TEM also accounted for most of the CD4^+^ T cells expressing TIGIT in the CD4_med_ and CD4_low_ groups. For CD8^+^ T cells, the majority of TIGIT^+^ cells were TEM and TEF.

For TIM-3, no differences were observed between the groups, and only low frequencies of TIM-3^+^ cells were observed in all the evaluated subsets (Figure 1c). Boolean analysis for the evaluation of the co-expression of exhaustion markers indicated higher frequencies of CD4^+^ T cells co-expressing TIGIT and PD-1 in all the PLWH groups (*p* = 0.039 for CD4_high_ and *p* < 0.0001 for CD4_med_ and CD4_low_) (Appendix A). Among CD8^+^ T cells, significantly higher frequencies of PD-1^+^TIGIT^+^ T cells were observed for CD4_med_ (*p* = 0.014), while a tendency of higher frequency was observed for CD4_low_ (*p* = 0.094). Significantly higher frequencies were also observed for all the PLWH groups when evaluating CD4^+^ T cells co-expressing three exhaustion markers (*p* = 0.011 for CD4_high_; *p* = 0.086 for CD4_med_; and *p* = 0.0069 for CD4_low_).

We also evaluated the frequency of general T cell subsets. As expected, the CD4_med_ and CD4_low_ groups presented a tendency of lower median frequencies of CD8^+^ TN cells, and the CD4_med_ group presented a tendency of higher median frequencies of TEM in comparison to HIV_neg_ (Appendix A).

In addition to an increase in exhaustion, we also observed elevated levels of CD4^+^ activation in the CD4_med_ (*p* = 0.021) and CD4_low_ groups (*p* = 0.0072) compared to HIV_neg_ (Figure 2). For CD8^+^ T cell activation, a significantly higher frequency was observed for CD4_low_ (*p* = 0.038), while CD4_med_ presented a tendency of higher frequencies of CD38^+^HLA-DR^+^ cells (*p* = 0.056).

Finally, we did not observe significant effects for any of the PLWH groups in the adjusted linear models used to evaluate senescent T cells (Figure 2), although the CD4_low_ group presented a significantly higher median frequency of CD8^+^CD28^−^CD57^+^ cells compared to HIVneg.

### 3.3. Immunophenotyping of Monocytes and B Cell Subsets

In addition to T cells, we aimed to investigate the subset profiles of monocytes as the balance between classical, intermediate, and non-classical monocytes is affected in HIV infection and correlates with an inflammatory setting in PLWH [32,33,34]. We observed a similar frequency of classical, intermediate, and non-classical monocytes between the PLWH and HIV_neg_ groups (Appendix A).

As for T cells, we also evaluated the frequency of naïve, activated memory (AM), resting memory (RM), and tissue-like memory (TLM) B cell subsets since the balance between those cells is impacted by HIV infection and because they drive humoral response. In addition, we also investigated the frequency of peripheral T follicular helper cells (pTfh, CD4^+^CD45RA^−^CXCR5^+^) as this subset has an essential role in the generation of high-affinity antibodies, memory B cells, and long-lived plasma cells [61]. The frequency of pTfh cells exhibited no significant differences between the groups, regardless of HIV status or CD4 count (Figure 3a). According to the multivariable analyses, all the PLWH groups presented significantly lower frequencies of RM B cells compared to HIVneg (*p* = 0.0026 for CD4_high_; *p* = 0.0004 for CD4_med_; and *p* < 0.0001 for CD4_low_), while increased frequencies were observed in the TLM subset for CD4_med_ (*p* = 0.028) and CD4_low_ (*p* < 0.0001) and in the AM subset for the CD4_low_ group (*p* = 0.03).

In addition to naïve and memory subsets, we also evaluated the frequency of total B cells, immature B cells, and plasmablasts (Appendix A). From these populations, a significant effect of a higher coefficient was observed for HIV_med_.

### 3.4. Higher Level of CXCL10/IP-10 in PLWH with Lower CD4 Counts

In addition to the frequency of cellular subsets, we also evaluated the plasmatic concentration of sCD163 and six cytokines that are consistently associated with inflammation or the regulation of the humoral response upon the activation/inhibition of B cells. Adjusted models using HIVneg as a reference group indicated higher concentrations of CXCL10/IP-10 in the CD4_low_ group (*p* = 0.013) and sCD163 in the CD4_med_ group (*p* = 0.023) (Figure 4). IL-18 values were similar among the groups, and most samples presented levels of IL-4, CCL4, IL-6, and IL-21 below the limit of detection.

### 3.5. YF-NAbs Are Lower in PLWH with Lower CD4 Counts and Correlated Negatively with T Cell Activation and Exhaustion

At D30, the HIV_neg_, CD4_high_, and CD4_med_ groups presented median titers of 1458, the upper limit of the neutralization test. For these groups, more than 70% of the individuals presented titers above 1:1000 (14/27 from HIV_neg_, 19/31 from CD4_high_, and 13/20 from CD4_med_). For the CD4_low_ group, we observed lower immunogenicity for the YF vaccine as the median YF-NAb titer in this group was 812, and linear model analysis showed a significantly lower linear coefficient for the group (*p* = 0.005, Figure 5).

Correlations between the D30 and D365 YF-NAb titers and the pre-vaccination levels of exhaustion, activation, and inflammation markers evaluated here were calculated (Figure 6). We observed a negative correlation between the frequency of exhausted CD8^+^ T cells (PD-1^+^, *p* < 0.028) and activated CD4^+^ T cells (CD38^+^HLA-DR^+^, *p* < 0.038) with the μPRN titers at D365.

## 4. Discussion

YF vaccine is one of the most effective vaccines currently available. Safety and efficacy have been extensively studied for the general population [21,62,63,64]. However, controversial data suggest that vaccine efficacy is impaired in PLWH [11,13,17,18,19,20], and the associated mechanisms are not clear. In a previous study [18], our group demonstrated lower YF vaccine immunogenicity in PLWH with low CD4 counts. However, the HIV-driven immunodysregulation associated with this reduced response was not evaluated. In the present study, we conducted an extensive immunological analysis to identify factors associated with lower YF vaccine immunogenicity in PLWH with low CD4 counts. We observed higher T and B cell activation and exhaustion in the PLWH groups with lower CD4 counts compared to the HIVneg group. Additionally, the PLWH group with lower CD4 counts also showed higher concentrations of IP-10, sCD163, and IL-18 compared to the HIVneg group [18]. We also observed a negative correlation between the pre-vaccination level of CD8^+^ T cell exhaustion and CD4^+^ T cell activation with YF-NAb titers at D365 and the pre-vaccination level of IP-10 with YF-NAb titers at D30 and D365.

The immune deregulation in HIV infection is driven by multiple factors, with generalized and chronic immune activation being considered key factors [23]. Here, we observed higher frequencies of activated T cells in the groups with lower CD4 ranges (CD4_med_/CD4_low_). Additionally, this suggests that, although more subtle than observed in other studies, our categorization of PLWH based on CD4 ranges allowed us to observe that these participants present different stages of immunological impairment, which could impact the vaccine response.

In a previous study by Muyianja et al. in 2014 [59], a negative effect on the magnitude of humoral/cellular response against the YF vaccine was observed in HIV-negative individuals with higher baseline activation and exhaustion markers. This study underscored the impact of immune activation on YF vaccine immunogenicity. In agreement with Muyianja’s findings, we observed a negative correlation between the frequency of activated CD4^+^ T cells with YF-NAb one year after vaccination.

A clear effect of immune impairment in PLWH observed in our study was an increased frequency of exhausted T cells compared to HIV-negative individuals, particularly in participants with lower CD4 counts. T cell exhaustion is characterized as a dysfunctional state driven by persistent antigenic stimulation, resulting in decreased proliferative capacity for general T cells, lower cytotoxic potential for CD8^+^ T cells, and lower polyfunctionality for CD4^+^ T cells [35]. In our study, we observed a profile of exhausted T cells in the CD4_med_ and CD4_low_ groups as participants from these groups presented higher frequencies of cells expressing PD-1 and TIGIT.

PD-1 is an inhibitory receptor that regulates T cell effector functions during various events, including acute and chronic infection, cancer, and immune homeostasis. In our study, we observed higher exhaustion in CD4^+^ T cells, with increased frequencies of PD-1^+^ T cells in total and TCM subsets for CD4_med_ and CD4_low_ compared to HIV_neg_. For the CD8^+^ T cells, increased medians were observed for the CD4_low_ group, but significance in the multivariable models was achieved only for the CD4_med_ group. This result confirms that, while CD8^+^ T cell cytotoxicity is crucial for the control of viral infections, exhaustion profiles also significantly impact CD4^+^ T cells and should widely impact the humoral immune response as these cells present an important role in antibody affinity maturation and the development of B cell memory.

While PD-1 plays a central role in the regulation of T cell responses and is considered a main exhaustion marker, several other co-inhibitory molecules are overexpressed in the context of T cell exhaustion. Among them, TIGIT and TIM-3 are widely co-expressed with PD-1, and their combination blockade leads to an increase in cytotoxic and antitumoral activity [40,65,66]. In our study, the frequencies of TIM-3^+^ cells were very low in all T cell subsets observed for all the groups, although elevated levels were described for T and NK cells in PLWH [41,67]. However, elevated TIGIT expression was observed even for the CD4_high_ group. These results agree with previous data that show that the exhaustion profile is not fully reversible in response to ART [27,68,69].

A negative effect of the increased expression of PD-1 was also observed by Muyianja et al. in 2014, with negative correlations between YF-NAb titers and the frequency of PD-1^+^ CD8^+^ memory T cells. Here, we also found a negative correlation between the frequency of PD-1^+^ CD8^+^ T cells before vaccination and the YF-NAb titers one year after vaccination, highlighting the impact of T cell exhaustion in antibody response.

In addition to T cell exhaustion, chronic immune activation accelerates T cell immune senescence, a profile associated with aging that affects the renewal capability of those cells and increases the secretion of pro-inflammatory cytokines [70]. In our study, we observed a significantly higher frequency of CD28^−^CD57^+^ T cells in the CD4_low_ group compared to HIVneg. However, we did not observe a correlation between senescence and NAb titers.

The CD4^+^ T cell is the primary cell subset impacted by HIV infection, but HIV-associated immune deregulation impacts other cell populations that are important for immune response. Monocytes, for example, are impacted by immune activation, and HIV infection favors differentiation towards CD16^+^ populations as intermediate and non-classical subsets [32,33,34]. In our study, however, the frequencies of the three monocyte subsets evaluated were similar in the PLWH and HIVneg groups and were not correlated with YF-NAb titers.

On the other hand, in our cohort, we observed the effects of immune activation in B cell subsets. As for T cells, B lymphocytes in PLWH present a shift towards more differentiated phenotypes in response to chronic stimulation of the immune system [49,50,51,52,53]. This shift was reflected in significantly higher frequencies of tissue-like memory B cells in the CD4_med_ and CD4_low_ groups. These cells are present at higher frequencies in PLWH [49,51,53,71,72] and are of particular importance as their profile resembles the one observed for exhausted T cells [51]. In this context, the increased frequency of these cells in individuals with lower CD4 T cell counts is another indicator of humoral response impairment in comparison to HIV-negative individuals or treated individuals with higher CD4 counts. Associations between frequencies of these cells and lower YF-NAb titers were also observed in the study by Muyianja et al. in 2014, but here, we did not observe a negative correlation between the frequency of TLM B cells with YF vaccine immunogenicity.

In addition to those observed in TLM B cells, we also observed lower frequencies of RM B cells in all the groups of PLWH, despite no significant alterations in the frequency of naïve or AM B cells. RM B cells are also named classic memory B cells as they represent the main circulating B cell subset [49], and several studies highlight that those cells are essential for the humoral response as they are long-lived memory cells with increased responsiveness compared to naïve B cells [73,74] and do not need continuous stimuli for long-term survival [73,75]. In HIV infection, however, these cells do not only occur in lower levels [52,53,71,76] but also present signals of dysfunctionality associated with impaired humoral response in viremic individuals [76,77,78]. Although B cell dynamics restoration was observed to some extent as a response to ART [52,78,79], we did not observe this effect in our cohort as participants from the CD4_high_ group presented RM B cell frequencies similar to the other PLWH groups. This should be reflected in an impaired humoral response as the preservation of this subset is associated with better immunological responses against HIV [80,81], similar to those observed for T central memory cells [30,31].

Although we evaluated an extensive panel of cells and molecules, our study presented some limitations. One of these limitations is the lower number of individuals in the CD4_low_ group and the absence of individuals without ART and a detectable viral load. The absence of individuals with detectable viremia in our cohort precluded the evaluation of the direct influence of HIV replication in vaccine response. Also, our study evaluated the YF-NAb titer only until one year after vaccination. Our cohort is still being followed, and the results from five and ten years after vaccination will allow us to evaluate if PLWH will maintain protective YF-NAb titers for a long time or if they will need a second dose of vaccine. Moreover, the limit of quantification of our YF-NAb assay probably impacted our correlation analyses when other studies evaluated higher YF-NAb titers.

## 5. Conclusions

We highlighted that immune activation, exhaustion, and inflammation persist in a group of PLWH with low CD4 counts, even with a suppressed viral load. These alterations could contribute to the diminished vaccine response observed in this group, as evidenced by the negative correlation between markers of activation, exhaustion, and inflammation with the YF-Nab titers after 30 days and one year after vaccination. Therefore, implementing strategies to decrease immune activation, exhaustion, and inflammation before vaccination could enhance vaccine immunogenicity in this population.

## Figures and Tables

**Figure 1 vaccines-12-00578-f001:**
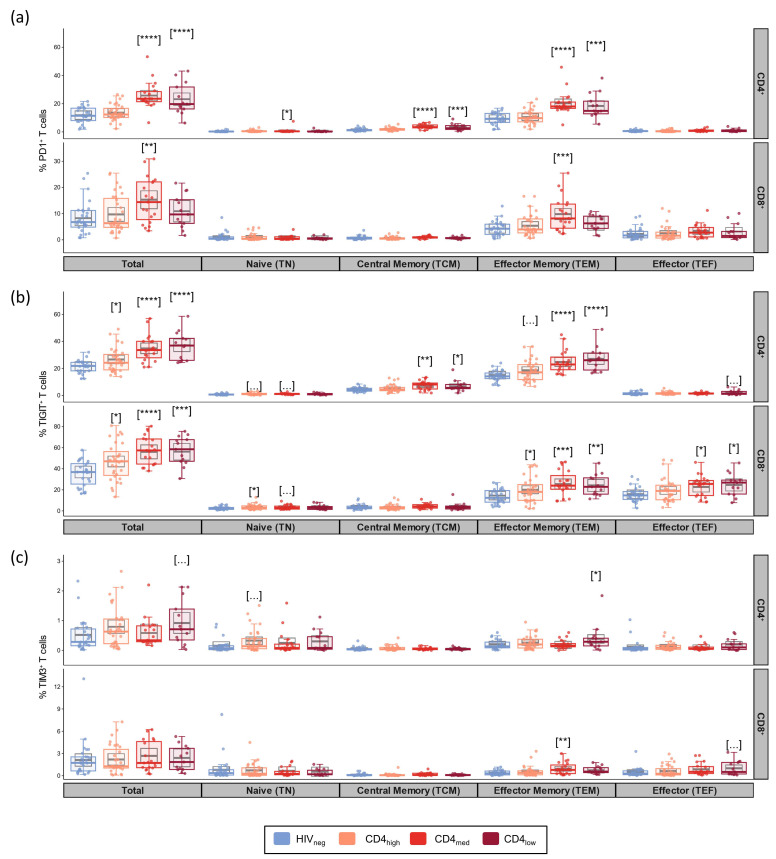
Exhaustion markers in total T cells and their subsets in PLWH and control groups. Frequencies of PD-1+ (**a**), TIGIT+ (**b**), and TIM-3+ (**c**) cells among CD4^+^ and CD8^+^ total, naïve (TN; CD45RA+CCR7+CD27+), central memory (TCM; CD45RA-CCR7+CD27+), effector memory (TEM; CD45RA-CCR7-CD27-), and effector (TEFF; CD45RA+CCR7-CD27-) subsets. The colored dots and boxplots are colored for each group evaluated according to the legend. For the colored boxplots, the horizontal bars represent the IQR and sample median, and the whiskers extend until the lower and upper fences. Gray boxplots represent marginal mean estimates and confidence intervals calculated for multivariate linear models fitted by ordinary least square regressions. *p*-values represent coefficient significance for the group in the calculated model using HIVneg as a reference and are represented between brackets as: … *p* < 0.1; * *p* < 0.05; ** *p* < 0.01; *** *p* < 0.001; and **** *p* < 0.0001.

**Figure 2 vaccines-12-00578-f002:**
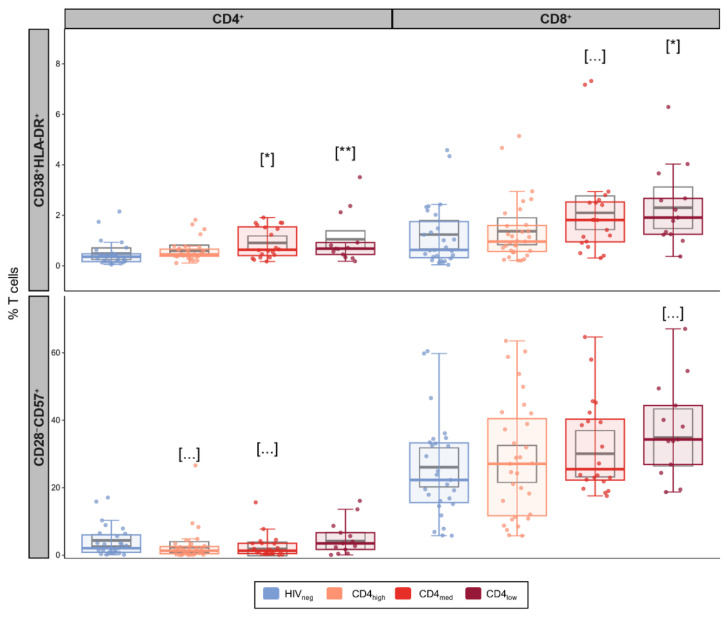
Senescence and activation in T cells in PLWH and control groups. Frequencies of activated (CD38+HLA-DR+) and senescent (CD28-CD57+) cells among CD4^+^ and CD8^+^ T cells. The colored dots and boxplots are colored for each group evaluated according to the legend. For the colored boxplots, the horizontal bars represent the IQR and sample median, and the whiskers extend until the lower and upper fences. Gray boxplots represent marginal mean estimates and confidence intervals calculated for multivariate linear models fitted by ordinary least square regressions. *p*-values represent coefficient significance for the group in the calculated model using HIVneg as a reference and are represented between brackets as: … *p* < 0.1; * *p* < 0.05; and ** *p* < 0.01.

**Figure 3 vaccines-12-00578-f003:**
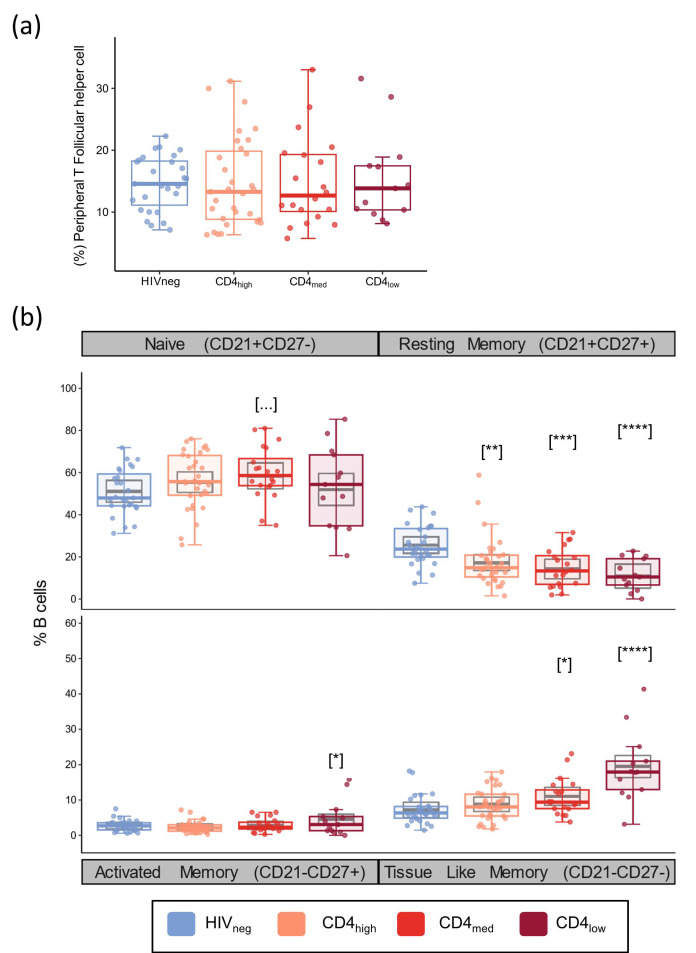
Peripheral T Follicular helper and B cell subsets in PLWH and control groups. Frequencies of peripheral T follicular helper cells (CD4^+^CD45RA^-^CXCR5^+^) (**a**), and naïve (CD21^+^CD27^-^), resting memory (CD21^+^CD27^+^), activated memory (CD21^-^CD27^+^), and tissue-like memory (CD21^-^CD27^-^) B cell subsets (**b**). The colored dots and boxplots are colored for each group evaluated according to the legend. For the colored boxplots, the horizontal bars represent the IQR and sample median, and the whiskers extend until the lower and upper fences. Gray boxplots represent marginal mean estimates and confidence intervals calculated for multivariate linear models fitted by ordinary least square regressions. *p*-values represent coefficient significance for the group in the calculated model using HIVneg as a reference and are represented between brackets as: … *p* < 0.1; * *p* < 0.05; ** *p* < 0.01; *** *p* < 0.001; and **** *p* < 0.0001.

**Figure 4 vaccines-12-00578-f004:**
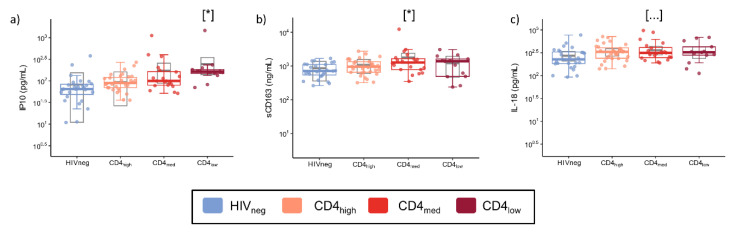
Plasma levels of inflammatory cytokines, cytokines involved in humoral response, and sCD163 in PLWH and control groups. The graphs represent the concentrations measured by a multiplex immunofluorescence assay of (**a**) IP10; (**b**) sCD163; and (**c**) IL-18. The colored dots and boxplots are colored for each group evaluated according to the legend. For the colored boxplots, the horizontal bars represent the IQR and sample median, and the whiskers extend until the lower and upper fences. Gray boxplots represent marginal mean estimates and confidence intervals calculated for multivariate linear models fitted by ordinary least square regressions. *p*-values represent coefficient significance for the group in the calculated model using HIVneg as a reference and are represented between brackets as: … *p* < 0.1; and * *p* < 0.05.

**Figure 5 vaccines-12-00578-f005:**
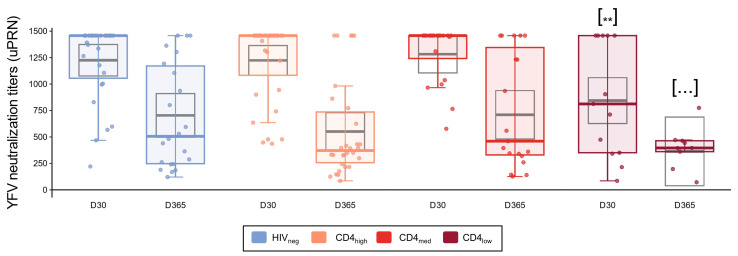
YFV neutralization titers in PLWH and control groups. YF neutralization antibodies titers at D30 and D365 as measured by the micro plaque-reduction neutralization-Horseradish Peroxidase test (YFV μPRN-HRP). For the colored boxplots, the horizontal bars represent the IQR and sample median, and the whiskers extend until the lower and upper fences. Gray boxplots represent marginal mean estimates and confidence intervals calculated for multivariate linear models fitted by ordinary least square regressions. *p*-values represent coefficient significance for the group in the calculated model using HIVneg as a reference and are represented as … *p* < 0.1; ** *p* < 0.01.

**Figure 6 vaccines-12-00578-f006:**
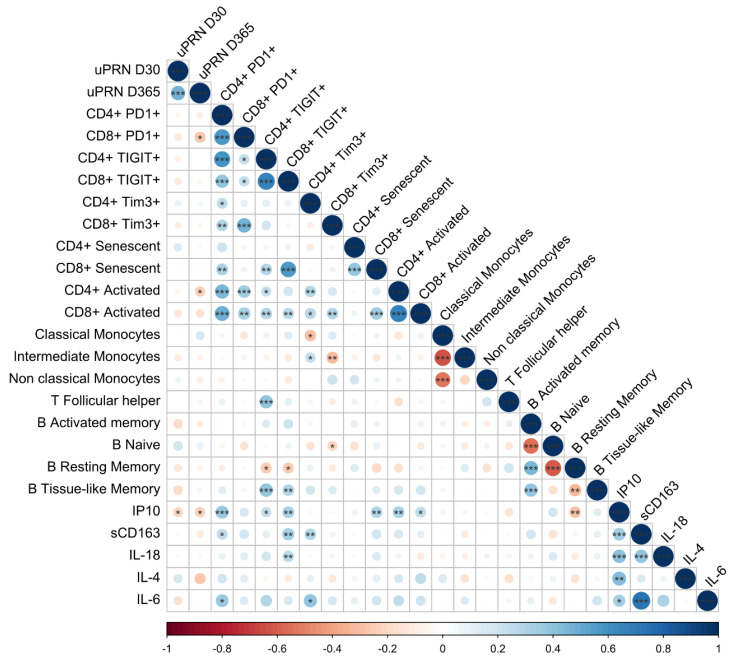
Spearman correlations between the markers evaluated in this study. Spearman correlation data are shown in the matrix and represent the correlation between the evaluated markers. Circles’ sizes and colors are equivalent to Spearman rho values obtained for each comparison. Significant *p*-values are represented inside the circles as * *p* < 0.05; ** *p* < 0.01; and *** *p* < 0.001.

**Table 1 vaccines-12-00578-t001:** Clinical and sociodemographic characteristics of the studied groups.

	HIV_neg_	CD4_high_	CD4_med_	CD4_low_	*p*-Value
	(n = 27)	(n = 31)	(n = 20)	(n = 13)	
**Sex [n (%)]**					
-Male	13 (48.1%)	21 (67.7%)	14 (70%)	10 (76.9%)	0.25
-Female	14 (51.9%)	10 (32.3%)	6 (30%)	3 (23.1%)	
**Age** **[median (Q1–Q3)]**	38.4 (28.94–47.01)	39.94 (34.78–46.08)	50.01 (39.29–52.76)	45.57 (37.35–49.46)	0.095
**Time since HIV-1 diagnosis [median of years (Q1–Q3)]**	N/A	9.4 (5.47–13.08)	9.43 (4.28–19.78)	2.38 (1.19–20.16)	0.562
**Nadir CD4** **[cells/mm^3^, median (Q1–Q3)]**	N/A	236 (126–352)	68.5 (29–124.25)	71 (59–123)	***p*** **< 0.001**
**Time on cART [median of years (Q1–Q3)]**	N/A	7.8 (1.77–8.98)	8.55 (3.84–15.52)	2.38 (1.04–20.16)	0.508
**CD4⁺ T cell counts** **[cells/mm^3^, median (Q1–Q3)]**					
-D0	1215.41 (958.32–1458.65)	730 (576–946)	396.82 (382.25–440)	296.62 (272.56–311.16)	***p*** **< 0.001**
-D30	1001.23 (771.08–1562.76)	701 (501.01–849)	387.65 (323.82–466.82)	333.79 (306.36–398.06)	***p*** **< 0.001**
-D365	1185 (980.79–1425.8)	710.96 (605.75–961.1)	536.62 (362.88–565.75)	365 (313.84–409.5)	***p*** **< 0.001**
**CD8⁺ T cell count** **[cells/mm^3^, median (Q1–Q3)]**					
-D0	590 (452.12–814.97)	828.52 (609.06–1354.74)	785 (537.83–1171.37)	1048 (598–1316.24)	**0.023**
-D30	562.92 (457.12–735)	835.69 (513.5–1113.37)	728.74 (536.29–1022.97)	913.11 (697.09–1276.9)	**0.007**
-D365	646 (403.5–811.14)	744.52 (579.5–1113.74)	1014.06 (881–1548.22)	1023.99 (970.24–1427.99)	**0.009**
**CD4/CD8 ratio** **[median (Q1–Q3)]**					
-D0	1.89 (1.39–2.4)	0.91 (0.64–1.22)	0.58 (0.34–0.82)	0.32 (0.23–0.42)	***p*** **< 0.001**
-D30	1.91 (1.35–2.34)	0.79 (0.61–1.25)	0.55 (0.36–0.77)	0.32 (0.27–0.51)	***p*** **< 0.001**
-D365	1.99 (1.5–2.53)	0.97 (0.68–1.18)	0.47 (0.38–0.53)	0.34 (0.3–0.42)	***p*** **< 0.001**

HIVneg—HIV-1-uninfected individuals; N/A—non-applicable; and Q1–Q3—1st quartile to 3rd quartile. *p*-values were obtained through a Kruskal–Wallis test or Fisher’s test, and *p*-values < 0.05 were considered significant and are highlighted in bold characters.

## Data Availability

Data supporting this manuscript may be available upon reasonable request to the corresponding author.

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
