# Peer review of "Impact of HIV-Related Immune Impairment of Yellow Fever Vaccine Immunogenicity in People Living with HIV—ANRS 12403"

_vaccines, 2024, doi:10.3390/vaccines12060578_

Round 1

Reviewer 1 Report

Comments and Suggestions for Authors

This paper presents a follow-up study of a previously published investigation of the immune response to yellow fever 17D vaccination in a cohort of subjects with HIV infection. YF vaccine can be given to HIV patients if their CD4+ cell count is >200 cells/mm3, however immunogenicity of the vaccine was shown to be impaired in these subjects especially in presence of HIV high viral loads or low CD4/CD8 ratio. This was confirmed also in the previous study of the authors. The focus of this paper is the evaluation of the baseline levels of several immunological markers pre-vaccination in a cohort of HIV-infected subjects in comparison to HIV-negative controls. The two groups showed the same decay of neutralizing antibody titer one year after vaccination however, significative differences were measured shortly after vaccination. In particular the group with the lowest CD4+ cell count also showed the lowest antibody titer. An increase in exhausted and activated T and B cells as well as the increase of inflammatory markers such as IP10 were correlated with the reduced response to the vaccine. This does not seem to be a specific feature of HIV-infected subjects because a similar profile has been described in other studies with HIV-negative subjects with elevated levels of activation and exhaustion markers.

The paper is well written, and the data are well presented. It underscores the importance of the baseline immune system activation level when a live attenuated vaccine is administered.

Major points:

The authors need to explain the rationale used in the separation of the HIV-infected patients into three groups based on pre-vaccination CD4 counts. They use three groups: CD4high (>500 cells/mm3), CD4med (250-499 cells/mm3), CD4low (200-349 cells/mm3) but the last two groups overlap.

The authors investigate the frequency of Tfh cells pre-vaccination finding no difference in the HIV-infected patients compared to controls. Since circulating Tfh cells have been shown to increase during YF17D vaccination, it would be interesting to check if the presence and magnitude of this increase in the presented study.

Minor points:

Line 97: The vaccine contains 10^5 particles not 105.

Line 353: remove “word”

Author Response

We thank the reviewer for the evaluation of our manuscript. Below, we present the point-by-point respnses regarding major and minor review points.

  • Major points:
  • The authors need to explain the rationale used in the separation of the HIV-infected patients into three groups based on pre-vaccination CD4 counts. They use three groups: CD4high (>500 cells/mm3), CD4med (250-499 cells/mm3), CD4low (200-349 cells/mm3) but the last two groups overlap.

We apologize, but the CD4 ranges were written incorrectly. CD4med group comprised individuals with CD4 counts between 350-499 and, therefore, none of the groups overlapped. The CD4 ranges used to separate groups were based on the WHO “Guideline on When to Start Antiretroviral Therapy and on Pre-Exposure Prophylaxis For HIV”, which considers CD4 counts above 500 cells/mm3 as normal counts for PLHIV and CD4 counts <350 cells/mm3 as indicative of advanced HIV disease.

 We corrected this information in the manuscript, lines 148-150: “CD4 counts >500 cells/ mm3 (CD4high; n=30), CD4 counts between 350-499 cells/ mm3 (CD4med; n=20), and CD4 counts between 200-349 cells/ mm3 (CD4low; n=13).”

  •  The authors investigate the frequency of Tfh cells pre-vaccination finding no difference in the HIV-infected patients compared to controls. Since circulating Tfh cells have been shown to increase during YF17D vaccination, it would be interesting to check if the presence and magnitude of this increase in the presented study.

 Thank you for the suggestion. We limited our experiments to pre-vaccination frequencies of the evaluated cellular subsets since we aimed to identify markers that could be informative of risk to lower vaccine immunogenicity before vaccination. In this context, altered cell dynamics post vaccinations were not in the scope of our work. Our group plans a future study evaluating YF-specific B cells responses and it will include Tfh cells evaluations.

  • Minor points

We thank you for the review and apologize for the errors. All minor points were addressed in the reviewed manuscript, as described below.

 Previous Line 97, now line 105: The vaccine contains 10^5 particles not 105.

 “105 viral particles.”

 Previous Line 353, now line 350: remove “word”

 “T cell exhaustion is characterized as a dysfunctional state driven by persistent antigenic stimulation”

Reviewer 2 Report

Comments and Suggestions for Authors

The authors present a study on the impact of HIV-related immune impairment in the yellow fever vaccine immunogenicity among people living with HIV.

From the introduction it is clear that yellow fever is a major concern for public health in certain regions and vaccination has to be done with precautions in immunocompromised individuals. Given that the YF vaccine is a live attenuated vaccine, it is expected that the vaccine can cause immunogenicity problems in immunocompromised people. It would be useful to mention the size of the population of PLWH in these regions, so to highlight the importance of this study.

The study design is described in the materials and methods session clearly, as well as the methods used for the analysis. In Table 1 authors listed the clinical and sociodemographic characteristics of the studied groups and in the HIV negative populations the male:female ratio was close to 50%, which is not the case in the PLWH. Does this influence the analysis?

The results are detailed, but the conclusion and discussion are not really clear to me and I do not see what additional conclusion can be drawn from these data compared to the one, where the data of this study was published (ref. 18). In case there is no additional information in this manuscript, I do not recommend it for publication. If there is an added value, I would recommend a major revision to make the discussion and conclusion clear for better understanding.

Comments on the Quality of English Language

Only minor editing is required, there are a few typos.

Author Response

We thank the reviewer for the evaluation of our manuscript. Below, we present he point-by-point responses about the questions raised.

  • From the introduction it is clear that yellow fever is a major concern for public health in certain regions and vaccination has to be done with precautions in immunocompromised individuals. Given that the YF vaccine is a live attenuated vaccine, it is expected that the vaccine can cause immunogenicity problems in immunocompromised people. It would be useful to mention the size of the population of PLWH in these regions, so to highlight the importance of this study.

Thank you for the suggestion. The epidemiological information about the Brazilian regions was added to the introduction, as described below.

Line 46-49: In Brazil, urban YF was eradicated in the late 1960s through vector control and vac-cination campaigns, but the disease remains endemic in the Amazon region [4], which extends through 9 states that house aproximately 15% of the country’s total population living with HIV, estimated at around 1 million [5].

Line 52: During the last decade, wide vaccination campaigns were called in response to out-breaks during 2014-2018 that reached the borders of greatly populated states in the southeast region [6,7]. Together, these states accounted for 162 thousand notified HIV cases in the last decade, representing approximately 40% of HIV infections in Brazil during the historical series [5].  

  • The study design is described in the materials and methods session clearly, as well as the methods used for the analysis. In Table 1 authors listed the clinical and sociodemographic characteristics of the studied groups and in the HIV negative populations the male:female ratio was close to 50%, which is not the case in the PLWH. Does this influence the analysis?

Thank you for the question. Although frequencies were not similar, the differences in sex between groups were not statistically significant by the Fisher exact test (p = 0.25). Despite that, we accounted for sex and age as possible confounding factors in our analyses, and these variables were considered as explanatory variables during statistical modeling. Due to this, we believe that possible sex differences did not influence our analyses

  • The results are detailed, but the conclusion and discussion are not really clear to me and I do not see what additional conclusion can be drawn from these data compared to the one, where the data of this study was published (ref. 18). In case there is no additional information in this manuscript, I do not recommend it for publication. If there is an added value, I would recommend a major revision to make the discussion and conclusion clear for better understanding.

We appreciate your comments. However, it is important to clarify that in the previous study (former ref 18, now ref. 22), reduced immunogenicity of the YF vaccine in PLWH with low CD4 counts was described without investigating the factors associated with this phenomenon. In the present study, we hypothesized that baseline immune deregulation caused by HIV infection before vaccination was associated with lower vaccine immunogenicity. We evaluated markers of immune activation, T cell exhaustion, monocyte subsets, B cell subsets, and inflammation to identify factors associated with lower YF-NAb titers. Our findings show that PLWH with CD4 counts <350 cells/mm3 had a higher frequency of activated and exhausted T cells and B cells than HIV-negative individuals. Additionally, we found a negative correlation between the pre-vaccination levels of CD8+ T cell exhaustion and CD4+ T cell activation with YF-NAb titers at D365 and the pre-vaccination level of IP-10 with YF-NAb titers at D30 and D365. We rewrite the first paragraph of the discussion to highlight the results from the current study, as described below.

 Line 325-335: “In a previous study [18][18], our group demonstrated lower YF vaccine immunogenic-ity in PLWH with low CD4 counts. However, the factors contributing to this reduced response were not evaluated. In the present study, we conducted an extensive immu-nological analysis to identify factors associated with lower YF vaccine immunogenicity in PLWH with low CD4 counts. We observed higher T and B cell activation and ex-haustion in the PLWH groups with lower CD4 counts compared to the HIVneg group. Additionally, the PLWH group with lower CD4 counts also showed higher concentra-tions of IP-10, sCD163, and IL-18 compared to the HIVneg group.[18][18] We also ob-served a negative correlation between the pre-vaccination level of CD8+ T cell exhaus-tion and CD4+ T cell activation with YF-NAb titers at D365 and the pre-vaccination level of IP-10 with YF-NAb titers at D30 and D365.”

 We also modified the conclusion to make it clear, as described below.

 Line 430-437: “We highlighted that the immune activation, exhaustion, and inflammation persist in a group of PLWH with low CD4 counts, even with suppressed viral load. These al-terations could contribute to the diminished vaccine response observed in this group, as evidenced by the negative correlation between markers of activation, exhaustion, and inflammation with the YF-Nab titers after 30 days and one year post-vaccination. Therefore, implementing strategies to decrease immune activation, exhaustion, and in-flammation before vaccination could enhance vaccine immunogenicity in this population.”

 Only minor editing is required, there are a few typos.

 We thoroughly reviewed the entire manuscript to address these issues.